# Transforming *Rhodotorula* sp. Biomass to Active Biologic Compounds for Poultry Nutrition

Daniela-Mihaela Grigore [1,2], Mădălina Ungureanu-Iuga [3,4,*], Elena Narcisa Pogurschi [1,*] and Narcisa Elena Băbeanu [2]

1. Faculty of Animal Productions Engineering and Management, University of Agronomic Sciences and Veterinary Medicine of Bucharest, 59 Marasti Blvd., District 1, 011464 Bucharest, Romania; daniela-mihaela.grigore@usamv.ro
2. Faculty of Biotechnologies, University of Agronomic Sciences and Veterinary Medicine of Bucharest, 59 Marasti Blvd., District 1, 011464 Bucharest, Romania; narcisa.babeanu@biotehnologii.usamv.ro
3. Mountain Economy Center (CE-MONT), "Costin C. Kirițescu" National Institute of Economic Researches (INCE), Romanian Academy, 49th, Petreni Street, 725700 Vatra Dornei, Romania
4. Integrated Center for Research, Development, and Innovation in Advanced Materials, Nanotechnologies, and Distributed Systems for Fabrication and Control (MANSiD), "Ştefan cel Mare" University of Suceava, 13th University Street, 720229 Suceava, Romania
* Correspondence: madalina.iuga@ce-mont.ro or madalina.iuga@usm.ro (M.U.-I.); elena.pogurschi@gmail.com (E.N.P.)

**Abstract:** In broiler chick-rearing, the color is usually acquired by synthetic carotenoids in addition to broiler diets (25–80 mg/kg feed), often represented by β-apo-8′-carotenal. In the past fifteen years, the demand for organic food products originating from free-range reared chicks started to grow, with a more directed awareness of the quality of meat and egg. Various investigations have been reporting microorganisms, such as the oleaginous red yeasts genus *Rhodotorula* sp., as fast-growing unicellular eukaryotes able to synthesize natural pigments. *Rhodotorula* sp. represents a perfect choice as a natural resource due to the capacity to adapt easily to the environment valuing low-cost sources of nutrients for their metabolism and growth. The biodiversity and the ecology effects establish novel boundaries regarding *Rhodotorula* sp. productivity enhancement and control of biological risks. It is, therefore, necessary to review the current knowledge on the carotenoid synthesis of *Rhodotorula* sp. In this paper, we aimed to address the pathways of obtaining valuable yeast carotenoids in different conditions, discussing yeast biosynthesis, bioengineering fermentative evaluation, carotenoid extraction, and the techno-economic implication of valuable pigment additives on poultry nutrition. Finally, the pro-existent gaps in research are highlighted, which may clear the air on future studies for bio-carotenoid engineering.

**Keywords:** artificial pigment alternative; broiler nutrition; carotenoids; health; pigment additives; vegetal waste

## 1. Introduction

Carotenoids are soluble pigments classified as tetraterpenoids divided as primary (hydrocarbons, carotene) and secondary as their oxidation product (xanthophylls). Widely, around 1100 different carotenoids [1] are synthesized in plant, algae, and fungi species. As natural lipophilic pigments [2], they are often characterized by a range of colors, starting from a pale and creamy yellow, light-pink, strong yellow, pink, and orange until strong red pigmentation and a rare, purple color [3]. Under natural circumstances, carotenoids have a multitude of roles, including sustaining photosynthesis, ensuring photoprotection [4,5], antioxidant capacity [6], reproductive enhancement [7], embryonal development [8], cell maturation [9], and immune system protection [10]. Birds cannot synthesize carotenoids hereby; carotenoids must be included in dietary intake. Dietary feed ingredients used in

commercial poultry feeding formulas are often processed in pelleted and extruded form, with consideration to nutrient availability and economic efficiency [11]. Mechanical procedures such as palletization or extrudate are currently employing high-temperature and pressure applied directly to feeding ingredients [12], thus affecting the retinol and retinol precursor by degradation, cumulating the vitaminic losses through handling and storage [13].

Carotenoids represent significant resources of retinol precursors (0.66 µg β-carotene = 1 U of retinol acid), with large implications for healthfulness and quality [14] products. Poultry is the most successful livestock sector around the globe and tends to grow due to the increasing consumption of poultry products. By the year 2020, the poultry sector generated almost 101 metric tons of meat and 1.65 billion eggs [15]. The great potential of carotenoid sources in industries (including food, feeds, nutritional supplements, pharmaceutics, and cosmetics) will have increased the forecasted market value to around $2.0 billion by 2022 [16]. The most commonly used food and feed colorant additives in poultry nutrition are xanthophylls (lycopene, canthaxanthin, astaxanthin, and zeaxanthin) that originate from almost 90% mainly synthetic resources. Annually, the market for pigment additives tends to grow by 8.2% percent during the forecasted period 2022–2032 [17] due to the increasing consumption of poultry products (meat and eggs). Currently, there has been a growing interest in obtaining organic pigment additives from non-conventional resources (algae, bacteria, and yeasts). The composition and the stability of the natural resources might be undefined and wide because of the complexity of biochemical metabolism and biological variability that is often associated with the cell structure. Great consideration was attributed to the carotenoid biosynthetic pathways of yeast, understanding the carotenoid yield, as productivity and integrity, with a view regarding product improvement and industrial scalability. In non-phototrophic microorganisms, carotenoids present a clear advantage in obtaining natural pigments [18]. One of the most important attributes is the capacity of microorganisms to use industrial waste as raw material substrate [19], hence increasing profitability and lowering the related costs of production. Many microorganisms synthesize carotenoids and present a valuable industrial potential (Table 1), although the data concerning *Rhodotorula* sp. yeast pigment application on livestock nutrition are few.

**Table 1.** Carotenoid-producing microorganisms.

| Microorganism | Carotenoid | Structure | Reference |
|---|---|---|---|
| Funghi | | | |
| *Neurospora crassa* | β-carotene |  | [20] |
| *Monascus* sp. | Monascorubramin |  | [21] |
| *Blakeslea trispora* | Lycopene |  | [22] |
| *Fusarium sporotrichioides* | Lycopene |  | [23] |
| *Aspergillus* sp. | β-carotene |  | [24] |
| *Pacilomyces farinosus* | Anthraquinone |  | [25] |

**Table 1.** *Cont.*

| Microorganism | Carotenoid | Structure | Reference |
|---|---|---|---|
| | | Bacteria | |
| *Paracoccus carotinifaciens* | Astaxanthin | | [26] |
| *Staphylococcus aureus* | Zeaxanthin | | [27] |
| *Zooshikella* sp. | Prodigiosin | | [28] |
| *Serratia marcescens* | Prodigiosin | | [29] |
| | | Yeast | |
| *Rhodotorula glutinis* | Torularhodin | | [30] |
| *Xanthophyllomyces dendrorhous* | Astaxanthin | | [31] |
| *Rhodototula mucilaginosa* | β-carotene | | [32] |
| *Saccharomyces neoformans* | Melanin | | [33] |

The current paper aims to highlight the multitude of approaches to obtaining valuable yeast carotenoids in different conditions, discussing yeast biosynthesis, bioengineering, fermentative evaluation, carotenoid extraction, and the techno-economic implication of valuable pigment additives on poultry nutrition.

## 2. *Rhodotorula* sp. General Aspects

The genus *Rhodotorula* sp. covers more than 165 species [34]. Morphologically, *Rhodotorula sp.* is a polyphyletic-shaped yeast [35] forming fast-growing colored colonies [36]. The proliferation of the *Rhodotorula* genus is generally regarded as asexual [37]; however, some strains belonging to the genus present sexual reproductive traits [38]. *Rhodotorula sp.* ecology and biodiversity cover a board of environmental varieties using a large variety of carbon resources, including glycerol [39], glucose [40], sucrose [41], galactose [42], and maltose [43], often encountered as dominant in yeast microflora (water, soil, vegetal, and animals) [44].

Yeast such as *Rhodotorula* sp. represents a perfect choice as a natural resource of secondary metabolites (Figure 1): carotenoids [45], lipids [46], and extracellular enzymes (Table 2). Saprophytic and ubiquitously found, the *Rhodotorula* genus possesses a full capacity for intracellular carotenoid biosynthesis [47] (provitamin A precursors, such as β-carotene and γ-carotenoid) [47,48], although the main carotenoids are torulene and torularhodin [49].

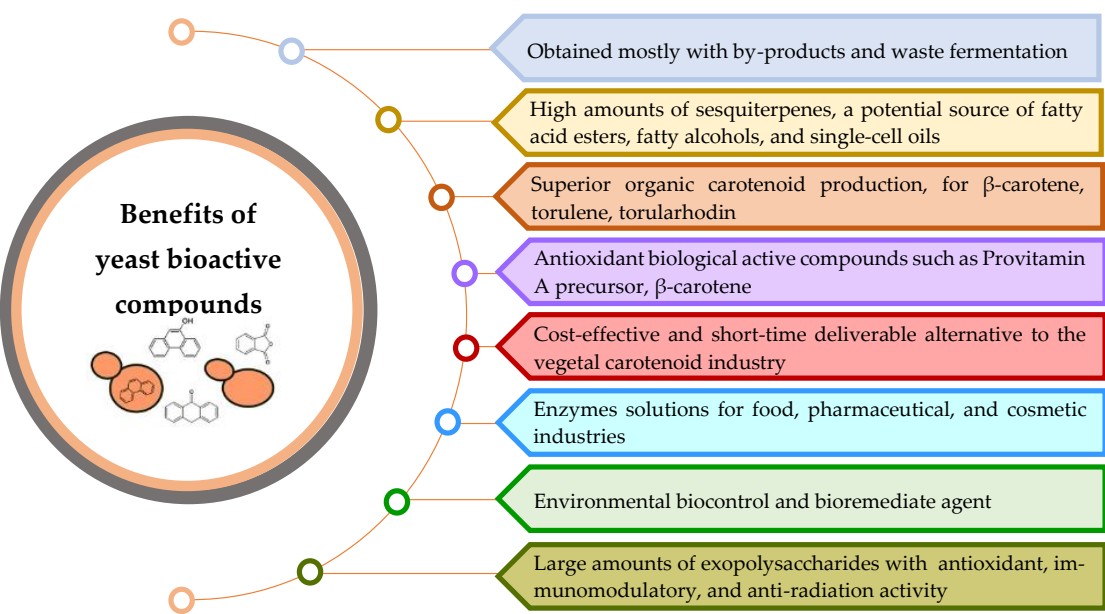

**Figure 1.** Major benefits of bioactive compounds from *Rhodotorula* sp.

**Table 2.** *Rhodotorula* sp. biomass active compounds.

| Yeast Strain | Assay Conditions and Results | Reference |
|---|---|---|
| | Oil←lipids→fatty acids | |
| *Rh. mucilaginosa* IIPL32 | Fed-batch C/N ratio = 40; Fed-batch C/N ratio = 60; Scalability: 50 mL→50 L; lipid yield for C/N ratio fed-batch = 40: 0.4 g→1.3 g/L lipid yield for C/N ratio fed-batch = 60: 0.45 g→1.8 g/L | [50] |
| *Rh. mucilaginosa* IIPL32 | (Lipids as FAME) 72 h→97.23 mg/g dry cell weight; 35–55%, MUFA C18:1 and C16:1 (oleic and palmitoleic acids) | [51] |
| *Rh. mucilaginosa* CCT3892 | The total amount of lipids obtained in the molasses medium was similar to the synthetic medium (15.36% ± 1.36% and 16.50% ± 0.68%, respectively), thus, the production of the metabolites was higher in the molasses medium. | [52] |
| | Enzymes | |
| *Rhodotorula mucilaginosa* CBMAI 1528 | Aspartic protease—pepsin family | [53] |
| *Rhodotorula mucilaginosa* | Invertase—the invertase with greater cell-structural stability and nystose productivity | [54] |
| *Rhodotorula* sp. Y-23 | Lipase (Lip-Y23)—low-temperature applications | [42] |
| *R. mucilaginosa* Y-1 | Carboxylase—Acetyl coenzyme A carboxylase (ACC1) | [55] |
| | Carotenoids | |
| *Rhodotorula glutinis* | β-carotene, torularhodin | [47,56] |
| *Rhodotorula mucilaginosa* KC8 | β-carotene, torularhodin | [57] |

Carotenoids are mainly synthesized via successive condensation (Figure 1) attributed to isoprenoid units such as isopentenic pyrophosphate (IPP) isomerized in dimethylallyl pyrophosphate (DMAPP) [4,58–63]. Particularly, yeasts such as *Rhodotorula* sp. possess the ability to transform lycopene into cyclic carotenoids like β-carotene (under lycopene β-cyclase action) and γ-carotene (conversion supported by lycopene cyclase).

The γ-carotene unit represents the main precursor for yeasts' carotenoid formations, as shown in Figure 2. β-carotene ($C_{40}H_{56}$) is the most common and abundant

precursor for retinol [64], strong-orange-red colored, chemically classified as isoprenoid (synthesized from eight isoprenoid units) [65]. Torularhodin is regarded as a xanthophyll ($C_{40}H_{52}O_2$—3′,4′didehydro β, ψ-carotene-16′oic acid) due to the presence of the carboxyl group [66] and represents the prevailing chemical structure in *Rhodotorula* sp. total carotenoid yield [67]. Torulene is classified as a carotenoid. The torulene molecule includes only hydrogen and carbon atoms [49], $C_{40}H_{54}$, 3′,4′ dihydro-β, ψ-carotene.

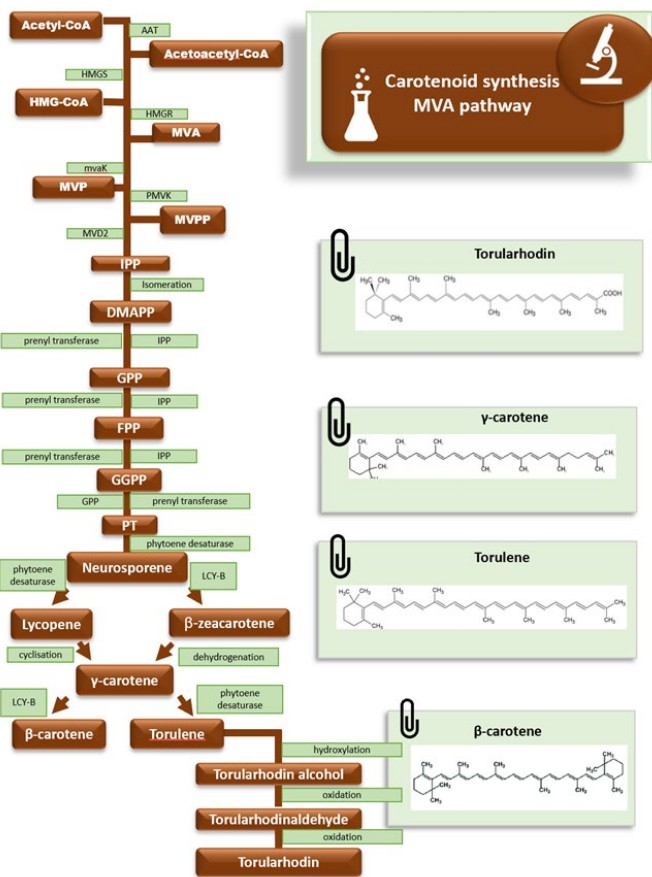

**Figure 2.** Rhodotorula carotenoid synthesis via MVA pathway; adapted after [68].

Concerning pigment bio-applications, *Rhodotorula* sp. yeast owns valorous advantages: fast-growing capacity, usually on organic waste materials (carbon-rich materials), and cost-effective production and harvesting (minimized human/process-associated interventions), which is more than suitable for large-scaling pigment industrial production and directly competing within the pigment market.

The vegetal sector is the most important resource concerning natural pigment industries. Factors such as poor soils [69] and climate-changing conditions [70] are currently affecting the industries, leading to long delays and negative economic implications within the pigment-related industries (food, feed, pharmaceutical, and cosmetic) [71]. Furthermore, the legislation and regulation of synthetic pigment use in food and feedstuff are narrowing the offer. Microbial pigment additives are still at the developing stage, up-scaling as the future organic source, an alternative to conventional resources, proving higher pigment capacity, in a shorter time. Recent studies regarding pigment-producing yeasts such as *Rhodotorula* sp. show an improvement in carotenoid productivity and componence that might be modulated [72] by mutagenesis [73] combined with other techniques, including different controlled stressors, such as temperature [74,75], substrate composition [76], lightning conditions [77] and aeration [78]. β-carotene, torularhodin, and torulene are valorous compounds synthesized exclusively by fungi and yeast [79], found in lipid bodies

of cellular biomass, and they are light unstable. The yeast carotenoid compounds present highly anti-oxidative characteristics [80].

## 3. Factors Affecting Pigmentation

### 3.1. Yeasts Nutrition

The productivity efficiency is generally based on the interdependency of the specific pathway involved and the culture medium (organic or inorganic compounds), strain specificity, and the growing conditions [81]. It is essential to balance the yeast requirements and cultivation conditions, to support the carbon-efficient use and yeast growth rates, thus optimizing secondary metabolites yields [46]. Moreover, individual metabolites yield (pigment quality and quantity) depends on the enzymatic complex employed in the yeast metabolic system [82]. Carbon and nitrogen have been considered the main sources of energy [83], and growth support of microorganisms, metabolite enhancement, and carotenoid producibility could be modulated by balancing the substrate composition, although, there are studies that show the induced stress by nutritive limitation (nitrogen sources) enhances carotenoid productivity (up to 0.75 mg/g dry cell weight) in Rhodotorula toruloides by suppressing the cell growth [84].

The influence of nutritional carbon/nitrogen resources on yeast fermentation and carotenoid yield might differ (Table 3) under the strain pigmentation capacity. Moreover, the nutritional carbon source might modulate the yeast carotenoid profile. Li et al. [83] showed that a medium containing glucose had more than 93% torularhodin in the total yeast carotenoid profile. There are also differences between the utilization of organic and inorganic compounds. Organic nitrogen resources such as yeast extract improve carotenoid yield up to 987 g/L [83], although using residual waste such as fruit and vegetable pulp/peel [85] and beer sludge represent hardly recycled market resources in developing countries, presenting worthy yeast nutritional potential.

**Table 3.** Factors affecting the carotenoid metabolism in yeast.

| Species | Factors Affecting Pigmentation Capacity and Productivity | | Results | References |
|---|---|---|---|---|
| | Mutagenesis | | | |
| *R. toruloides* NP11 | Atmospheric exposure, 30 °C and plasma technique followed by chemical mutagenesis with nitrosoguanidine | | *R. toruloides* XR-2, colonies of dark-red colored Nitrogen limitation conditions induced slower growth with high carotenoid yields | [84] |
| *Rhodotorula* sp. Strains | T-DNA insertional mutagenesis gene discovery in *R. toruloids* | | Twenty-seven mutant yeast phenotypes for lipid and carotenoid metabolism | [81] |
| *R. toruloides* NP11 | An *Agrobacterium tumefaciens*-mediated transformation (ATMT) to change its carotenoid production and profiles | | Selected three new phenotypes and mutants with different colors Characterized their carotenoid products | [86] |
| *Rhodotorula toruloides* CBS 14 and NCYC 1585 | Cloning strategy; four inducible promoters for control gene expression in *Rhodotorula toruloides* to obtain molecular genetic tools for manipulation | | Directed genetic and gene expression for carotenoid and lipid yields in *Rhodotorula toruloides* | [87] |
| | Cultivation medium | | | |
| | Carbon | Nitrogen | | |
| *Rhodotorula* sp. | - | Threonine (0.1, 0.2, 0.3%) glutamic acid (0.1, 0.2, 0.3%) | Both amino acid stimulation enhanced yeast growth parameters and total carotenoid formation | [76] |



**Table 3.** *Cont.*

| Species | Factors Affecting Pigmentation Capacity and Productivity | | Results | References |
|---|---|---|---|---|
| *Rhodotorula mucilaginosa* | Waste from the olive oil industry (Alperujo water, AE) in different aqueous solutions at concentrations: of 5, 10, 20, and 30% | | The volumetric carotenoid production significantly increases in 20 and 30% AE concentration (up to $7.3 \pm 0.6$ mg/L total carotenoids) | [67] |
| *Rhodotorula* sp. RY1801 | Sucrose, lactose, maltose, fructose and glucose | Inorganic nitrogen: ammonium sulfate, ammonium nitrate; Organic nitrogen: yeast extract, urea | Carbon sources: glucose; carotene yield up to 962 µg/L Nitrogen sources: yeast extract; carotene yield up to 987 µg/L | [83] |
| *Rhodotorula glutinis* (AS 2.703) | - | Peptone (PEP), yeast extract (YE), and ammonium sulfate | The highest biomass accumulation was 12.2 g/L after 144 h (YE) | [88] |
| *Rhodotorula mucilaginosa* | Onion and potato (skin) | Mung bean (husk) and pea (pods) | The highest carotenoid yield was archived by using onion peel extract and mung bean (up to 717.82 µg/g) | [79] |
| Lightning conditions | | | | |
| *Rhodotorula mucilaginosa* | Irradiation UV-C—254 nm | | Metabolite production by psychro-tolerant *Rhodotorula mucilaginosa* produced up to $56.9 \pm 3.2$ (µg/g$^{-1}$.dry weight) of total carotenoids | [89] |
| *R. glutinis* (CGMCC No. 2258) | LED lamp's light exposure intensities (4000 lx and 8000 lx) | | The lipid and β-carotene production enhancement by using light exposure and sodium acetate componence substrate in *R. glutinis* | [56] |
| *R. mucilaginosa* K-(1) | Two lighting shakers: Shaker 1: 1700 lx; Shaker 2: 3500 lx. Settled as 12 h dark: 12 h light | | Illumination intensity increases the carotenoid yield (1700 lx); High illuminating intensity (3200 lx) inhibits yeast glucose metabolism. Thus, cell growth | [77] |
| *R. glutinis* (CGMCC No. 2258) | Two groups: without and with continuous irradiation (3400 lx) | | Continuous irradiation might positively affect the lipid and carotenoid content | [90] |
| *Rhodotorula mucilaginosa* | Stress conditions: ultraviolet (UV) light and photoperiods | | Optimum conditions for stimulating the carotenoid productivity were 1 min of UV exposure combined with 0.5 mg/L magnesium sulfate and 18:6 h lighting conditions | [91] |
| Thermic conditions | | | | |
| *Rhodotorula* sp. RY1801 | Incubation temperature ranges from 20 to 37 °C | | Optimum incubation temperature at 28 °C | [83] |
| *Rhodotorula glutinis* | Incubation temperature ranges from 25, 30, 35 to 40 °C | | Optimum incubation temperature at 30 °C | [92] |
| *Rhodotorula mucilaginosa* | Incubation temperatures 15, 20, 25, 28, 30, 35, and 40 °C | | The most suitable temperature for culture growth and carotenoid production was 28 °C | [93] |

**Table 3.** *Cont.*

| Species | Factors Affecting Pigmentation Capacity and Productivity | Results | References |
|---|---|---|---|
| *R. mucilaginosa* ATCC 66034 *and R. gracilis* ATCC 10788 | Incubation temperature 20 °C and 28 °C | Optimum incubation temperature at 20 °C | [39] |
| [74] | Low temperature (16 °C) treatment Control temperature (25 °C) treatment | At 16 °C, the carotenoid yield was significantly increased | [75] |
| Aeration conditions | | | |
| *Rhodotorula glutinis* NRRL Y-12905 | Different conditions of agitation (150 to 250 rpm) and aeration [(2.5 to 5.0 of flask volume-to-medium volume ratio (vvm)] | Agitation and aeration at 250 rpm and 5.0 optimal conditions (high yeast cell concentration) | [78] |
| *Rhodotorula rubra* PTCC 5255 | Aeration levels: 0.115, 0.345 and 0575 vvm | The optimum carotenoid concentration was found at an aeration rate of 0.469 vvm, having the substrate initial pH of 6.48, and light intensity of 1757.84 lx | [94] |
| *Rhodotorula mucilaginosa* MTCC-1403 | Different conditions of agitation: 80, 110, and 140 rpm | Elevation of up to 100 μg carotenoids per g of dry biomass | [79] |
| Metabolizable salts and microelements addition | | | |
| *Rhodotorula glutinis* CCT 2186 | Different experimental levels of: glucose, $KH_2PO_4$, $MgSO_4$, $NH_4NO_3$, and pH | Combined sources of inorganic and organic nitrogen sources had high productivity yields | [45] |

Sharma and Ghoshal [79] used onion peel, mung bean, and pea (agro-industrial wastes) as a substrate for pigment production on *Rhodotorula mucilaginosa*, obtaining the best carotenoid productivity (27.4 mg/L) on onion peel extract. The olive oil industrial waste (20%-culture media) improved the total volumetric carotenoid production (up to 5.5 g/L) [67]. Carrot peels or starch in the potato feed industry is a typical example of recoverable fractions either as solids or as sludge which, after drying and sterilization, can be included directly in yeast bioprocess as sources of carbohydrates [95]. By recycling vegetal waste and thus improving the culture media for yeast growth, the productive-related costs are reduced to a minimum or absent in the development of the market economy. The costs involved mainly in recovering profitable nutrients from food waste processing; the credit played is derived from nutritional applications useful in all agriculture branches. At the same time, the economy of waste conversion and valuable byproducts generates a new secondary-industry domain, with new jobs and skills at the place of production [96].

### 3.2. Yeasts Fermentation Conditions

Carotenoid yield related to obtaining secondary specific metabolites might be an induced response generated by different stressors applied to the *Rhodotorula* sp. growth (nutritive limitation, aeration, and temperature) and could influence (by delaying or accelerating) the carotenoid synthesis. It has been demonstrated that the yeast carotenoid yields maximum values within reaching the cell's mature development [97]. Furthermore, the variability of the carotenoid yield componence proportions within the mature cells variates depending on the temperature and time of cultivation, 144 h on yeast malt, 252.99 μg/g total carotenoids [98], 120 h on yeast malt, 223.5 μg/g total carotenoids production [99].

The temperature parameter is a critical cultivation factor in the first place, affecting culture viability and biomass productivity and active bio compounds quality. Temperature correlates with metabolic functions and influences enzymatic activity [100] with carotenoid productivity, hence, effective regulation between cyclic carotenoids synthesis,

followed by precursors. An indirect metabolic synthesis between the low temperature of β-carotene synthesis and the opposite [101], increasing xanthophyll's and β-carotene precursors concentrations at higher temperature values, is probably by the low-temperature enzymatic activity of lycopene β-cyclase. Recent research points out that higher values of β-carotene production were recorded-by at 20 °C 250 mg/L representing 92% of total carotenoids compared with 30 °C, 125 mg/L, and the amount of 60% from total carotenoids and 35 °C, with less than 19% β-carotene and torulene encountered in biomass; although at 35 °C the torularhodin synthesis increased, leading up to 78% of total carotenoids in biomass [29,39,102].

Yeasts such as *Rhodotorula* sp. have naturally developed a light-sensitive response to environmental lightning conditions, protecting the yeast cells by synthesizing a large amount of β-carotene. White light irradiating trials were conducted on 21 strains of *Rhodotorula* sp., and the results concluded that the amount of carotene is twofold higher by irradiation (14.2 mg/100 g dry weight biomass). At the same time, light irradiation as a photo-regulative measure could modulate yeast growth and biochemical componence to enhance carotenoid productivity [77], although strong light exposure could negatively affect the yeast cultures, inhibiting their growth.

*Rhodotorula* sp. is an oxygen-dependent yeast [103] affecting both viability and productivity. Recent studies regarding the oxygen demand have demonstrated that the yeast cell growth and metabolism are strongly crisscrossed with yeast phenotype and the yeast-applied stressors, confirmed by the secondary metabolite's yields [30] and other bioactive compounds such as hemoproteins [104].

Besides the photo-protective role, yeast carotenoid active compound has an oxidative protection function facing the oxidant agents before yeast cell wall attack [105]. Oxygen supply, through aeration, agitation, or airlift bioreactors, is crucial to yeast metabolite productivity. Yeast oxygen requirements concerning carotenoid productivity were studied, and the results show an increase of end-metabolites synthase (torularhodin) expected from cyclic carotenoid oxidation [106].

## 4. Yeasts Pigment Extraction and Quantification

Yeast carotenoid yield determinism is directly modulated by the yeast phenotype and the engineering approach via metabolites enhancement. Despite the progress achieved in the biotechnological yeast carotenoids synthesis optimization, there is a permanent need for research efforts to constantly adapt and improve the in-process efficiency and minimize the economic implication. The yeast fermentative process is followed for the quantification of productivity determinations: preparative (harvesting, cell biomass disintegration) and quantitation methods (extraction, separation, and evaluation).

Harvesting viable cells, carotenoid extraction and purification of the carotenoid components are the most expensive procedures in techno-economic analysis. There are many ways to process yeast carotenoids. Harvesting cell biomass can be easily achieved by mechanical, chemical, or biological strategies. The centrifuge separation is the conventional mechanical method used in yeast industries, employed at 8000–10,000 rpm during a period of 7–10 min [107–109]. Current innovative methods concerning biomass harvesting are flocculation, pre-concentration techniques, high-pressure filtration, flotation, osmosis, bubble columns, and exploitation of hydrophobicity/hydrophobicity yeast proprieties [110]. The appropriate harvesting method is generally chosen through yeast proprieties such as cell size, biomass density, production volume, and final product specificity. Consecutive in yeast recovery, the yeast cell purification techniques, as successive washing with solvent and filtration cycles with the purpose of cell biomass clear separation. There are many carotenoids extractive methods [30,111–113] for samples and pure specific carotenoid quantification. Microbial carotenoids are secondary metabolites [114], present in almost 95% of the cell. Cell wall disruption and disintegration are needed as preparative procedures in carotenoid extraction. The most common approach is an organic/solvent-free mechanical breakage [115,116] combination between sonication/pressure treatments or

freeze-thawing/sonication [117,118] without having major losses on the yeast cell biomass compared to the synthetic chemical disruption that might generate artifacts or radicals [119], artificial condensation (acetonides) [120] or at worst, generating radioactive components (aldehydes) [121]. There are cell breakage methods that are less harmful, involving hydrolysis, supercritical CO2 [122], or enzymatic digestion extraction [123,124], having superior recovery rates, and implying extra financial costs. Carotenoids are non-polar chemical compounds characterized by water insolubility. A more hydrophilic carotenoid form is represented by their derivates, xanthophylls, due to the hydroxyl radical on the chemical structure. The commonly used extractive processes imply the reagents (acetone, cyclohexane, dimethyl sulfoxide, chloroform, petroleum ether, and ethyl acetate) usage as extractive solvents to separate the pigment compound in the partitioned liquid of analysis [125,126]. Carotenoids and xanthophylls are chemical compounds having more than nine double bonds that are capable of light absorption, detected between visible/UV wavelengths range of violet and blue-green spectra (450–550 nm) [127], naturally reflecting red, orange, and yellow color shades. Carotenoid detection and quantification have various protocol approaches, employing spectrophotometry or spectroscopy determinations. Pigment quantification assays are practically based on a comparative determination against pure chemically carotenoid materials (commercially available standard references, as 95–99% pure, for specific determinations), lab standardized as etalons curves, as for accuracy, reproducibility, and repeatability (as for peaks, retention time and area of peak) that later on might be interpreted as values using conversion formulae [128]. Carotenoid UV-Vis assay is a feasible method of quantification but needs a long time to determine because carotenoids obey the law of Lamber–Beer (the compound concentration is directly proportioned with the compound spectral absorbance) [129]. The UV-Vis conducted assays evaluate the liquid carotenoid sample (up to 3.0 mL) compound against the pure carotenoid standard reference substance with the intention of total carotenoid measurement [130]. The disadvantage of employing the UV-Vis method is the mediocre specificity consisting of the incapacity of distinction between individual carotenoids (similarity of peaks and absorbance wavelength around 459–500 nm for more than four distinct carotenoids). Quantitation is possible only by mathematical determination by using specific carotenoid partition coefficients [131]. A more precise approach is using the HPLC method (high-pressure liquid chromatography). Despite the time and costs regarding reagents and capillary system components, carotenoids are detected and measured simultaneously and accurately quantified individually [132] needing no more than 1.5 mL of liquid sample, injected (40 μL) with high pressure, carried (flow rate: 0.5 mL/min) with the eluent (A: acetonitrile: water, 9:1 and B: 1% formic acid ethyl acetate) to the stationary component (column C18, 250 mm × 4.6 mm, 5 μm) and detected (UV detector) [88]. Moreover, it highlights the labor exercise that lies in the systematic examination of the spectral signature, which is no longer just that of the compound of interest, needing specific determination to identify and quantify impurity componence. The FTIR (Fourier transform infrared spectroscopy) is capable of simplifying the total carotenoid quantitation, not only by time (less than 150 s) and cost but also by accuracy, dividing them as chemical structures [133]. RAMAN spectroscopy is the superior method of determination, analyzing at the same time light absorbance and matter structure of the sample only by photon laser interaction with the small sample size, in a very short time determination—based on the relation of light interaction on all materials, scattering the same amount of energy as incidence light [134].

## 5. Yeast Carotenoids in Poultry Nutrition

### 5.1. Retinol Requirements and Retinol Precursors in Poultry

The challenge regarding poultry vitamin requirements is and will be an actual research domain due to the genetic abundance and oscillational nutritional aspects between various factors that appear in poultry-intensive sectors (health status, veterinary medications, feed, breed, age, housing aspects, and rearing technology). Poultry specialized hybrids have exigent vitamin A requirements (Table 4), solidly correlated with the breed's purpose and

rearing management recommendation. In poultry, both layers and meat broilers have an excessive level of vitamin A feed supplementation starting from 10,000 International Units (IU)/kg diet up to 13,000 IU/kg diet, according to supplier recommendation, despite the requirements profile established by the National Research Council (NRC, 1994) colorant additives should not exceed 4500 IU/kg-fed meat broilers and 2500 IU/kg-fed layers. Moreover, vitamin supplementation is recommended to be equal to or more than birds' requirements [135], hence avoiding vitamin deficiency. In poultry, provitamin A and retinol deficiency could be a consequence of malabsorption or the impossibility of metabolic conversion, often regarded as biologically available [136]. The effects concerning retinol deficiencies are complex and affect a large range of metabolic activities: weight loss cumulated with slowing down the growth processes and negative performance rates [137], follicular hyperkeratosis, epithelial lesions, xerophthalmia [138], keratomalacia, hemeralopia, reproductive system malfunction [139], and gastrointestinal disorders. Exceeding retinol and provitamin A in poultry leads to xanthomatotic disorders [140] and hypercalcemia, followed by bone system disorders. The vitamin A origin and the stability within the dietary intake is a current challenge, although most of the commercial feeding formulas are developed and balanced by adding artificial vitamins along with micro and macro elements and by not taking into account the vegetal raw material vitamin content, thus the vitamin antagonistic [141] or destructive compounds [142]. Additionally, naturally occurring vitamins in feed and forages are presenting stability issues [61] due to inadequate feed manipulation and storage, often causing vitamin oxidation and frequent bacterial infestation [143], implying constant economic depreciation and loss [144]. Furthermore, the treatments such as insecticides and pesticides administrated to livestock crops are interfering with and affecting the feed vitamin concentration, leading to toxic traces within grain cultures [145].

**Table 4.** Broiler chicks and laying hens' pigment additive (IU/kg fed) in dietary-fed formulas *.

| Broiler Chicks | | | | |
|---|---|---|---|---|
| **Hybrid** | **0–11 Days** | **12–23 Days** | **24–42 Days** | **References** |
| Cobb 500 | Up to13,000 | 10,000 | 10,000 | [146] |
| Ross 308 | 13,000 | 13,000 | 13,000 | [147] |
| Arbor acres | 13,000 | 10,000 | 10,000 | [148] |
| Hubbard | 13,000 | 13,000 | 13,000 | [149] |
| **Laying Hens** | | | | |
| **Hybrid** | **0–6 Weeks** | **7–12 Weeks** | **12–18 Weeks** | **>18 Weeks** | |
| Hy-line W36 | 5700 | 5700 | 5700 | 5700 | [150] |
| ISA chick | 15,000 | 15,000 | 13,500 | 13,500 | [151] |
| Lohmann | 10,000 | 10,000 | 10,000 | 10,000 | [152] |
| **Turkeys** | | | | |
| **Hybrid** | **0–42 Weeks** | **43–84 Weeks** | **>84 Weeks** | |
| Hybrid Grade Maker male turkey | 9000–12,000 | 9000–12,000 | 8000–11,000 | [153,154] |
| Hybrid meat turkey | 13,500 | 12,000 | 11,000 | [155] |
| **Ducks** | | | | |
| **Hybrid** | **Starter** | **Grower/Finisher** | | |
| Longyan laying ducks | 10,000 | 8000/12,000 | [156] |
| Pekin | 10,000 | 10,000 | [157] |

* As supplier nutritional guidelines.

### 5.2. Carotenoid Absorption in Poultry

The physiologic and biochemical roles of provitamin A and retinol precursor cover multiple functions. In poultry, in the starter growing phase, retinal and provitamin A stimulates the growth processes and normal development of the reproductive system [158]. Provitamin A is the most important in preventing epithelial disorders (conferring elasticity

and anti-infectious resistance) and maintaining homeostasis of the visual function [159]. In poultry physiology, carotenoid synthesis is absent. Therefore, an exogenous intake is required. Birds can synthesize retinol from β-carotene through the retinal enzyme [90,160], β, β-carotene-15,15′-monooxygenase, capable of separation into two retinal symmetrically molecules [161]. The vitamin A precursor, β-carotene is an indispensable nutrient for reproduction, growth, and production (the biological activity is almost 60% of retinol activity). β-carotene absorption and bio-disposable variates by the bird's metabolism, the bird's absorptive capacity, and the forage quality related to formula stability and biochemical characteristics. Physiologically (Figure 3), β-carotene is a long-term absorption compound (up to three days until retinol conversion) that combines into chylomicrons in the small intestine mucosa (duodenum) and is carried further to the liver through the portal vein [162]. Oil presence enhances the vitamin A precursors absorption and liver metabolization [163], combined with lipoproteins in triglycerides (VLDL and LDL) and transferred to a specific tissue (skin, meat, fat, ovary, and egg yolk). Retinol in excessive quantity is moreover deposited in the liver and blood, then in muscle, fat, eggs, or skin [164]. Egg yolks' carotene deposits vary between 40–50% of total carotenoid intake [165]. However, most of them are lycopene, canthaxanthin, astaxanthin, and zeaxanthin, and lidding the β-carotene yolk concentration less than 1% due to the higher xanthophyll absorption in the bird's digestive tract [166]. Moreover, the bioavailability of carotenoids is mostly influenced by the matricidal food structure, carotenoid compound chemical structure, and interaction with other dietary nutrients.

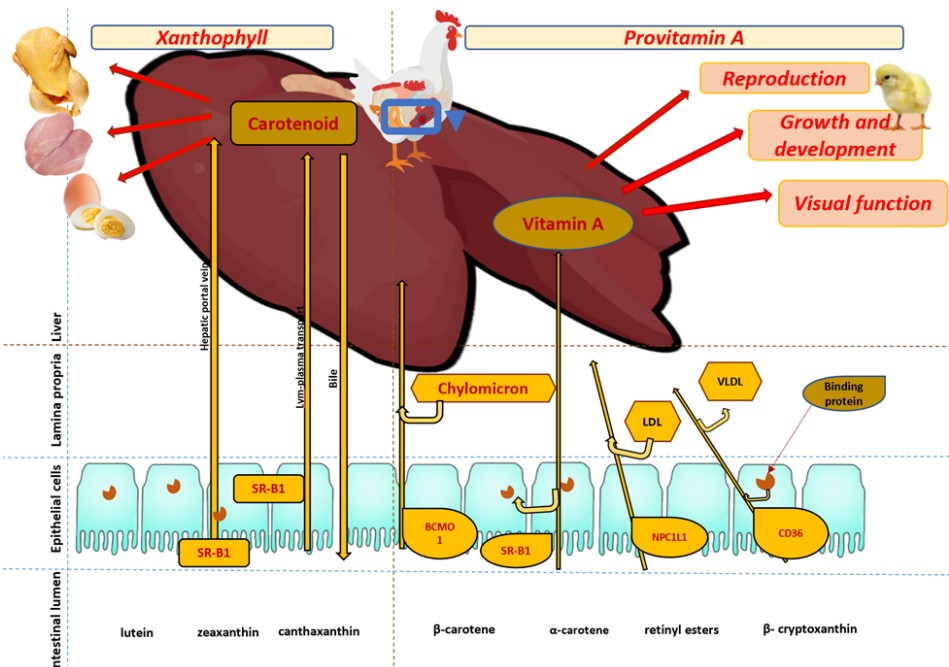

**Figure 3.** Carotenoids metabolism in poultry physiology; adapted after [167].

### 5.3. Poultry Feed Sensorial Additives

Nine pigment additives (Table 5) are regulated and used widely in the EU (indexed as pigment additives E160, E161, E162, and E163) [168] as appropriate for poultry nutrition labeled (EU Council directive 96/23/EC, 2010) as dietary pigment additives (SafeFood, 2022), for improving the egg yolk skin and meat product color.

**Table 5.** Pigment additives used in poultry nutrition [1].

| Pigment Additive [1] | Code [2] | Meat * | Egg * | Origin |
|---|---|---|---|---|
| β-Apo-8′-carotenal | E 161e | 80 | 80 | artificial |
| Cryptoxanthin | E 161b | 80 | 80 | natural |
| Lutein | E 161b | 80 | 80 | natural |
| Ethyl ester of β-apo-8′-carotenoid acid | E 160f | 80 | 80 | artificial |
| Zeaxanthin | E 161h | 80 | 80 | natural |
| Violaxanthin | E 161e | 80 | 80 | natural |
| Citranaxanthin | E 161i | - | 80 | artificial |
| Canthaxanthin | E 1601g | 25 | 8 | artificial |
| Capsanthin | E 160c | 80 | 80 | natural |

[1] Pigment additives regulation in poultry feed approved by the E.U.; [2] additives encoding by Council directive 70/524/EEC; * expressed as mg/kg diet.

The use of commercial carotenoids in poultry feed formulations is expensive and originates from around 90% artificial sources. Most of the pigment additives are approved for dietary inclusion up to 80 mg/kg in broiler chicks feeding formula [11], except canthaxanthin. Canthaxanthin levels are restricted and should not exceed more than 8 mg/kg for laying hens and 25 mg/kg feed for broiler chicks [169]. As the sole pigment additive used in human, fish, and poultry nutrition, canthaxanthin [170] dietary overdosage leads to residual pigment tissue deposits that expose the final consumer to a pigment intake that exceeds the Acceptable Daily Intake (ADI, 0.03 mg/kg body) [171], and might negatively affect the consumer's health (high risk of toxicity). Research concerning natural carotenoid sources as an alternative to commonly used synthetics for livestock nutrition shows that using natural resources such as maize and pasture (fresh or preserved) [172,173] and genetically modified organism (GMO) or non-GMO (plants, algae, and yeasts) could serve as superior native carotenes used pigment additives [174–176]. Few studies regarding the microbial piments additives on broiler meat [45,177,178]. Dietary inclusion of red yeast *Phaffia rhodozyma* (10–20 mg/kg feed) on broiler chicks positively affected the broiler chicks' performances and immune response, presenting 10 times stronger pigment capacity [179]. Moreover, in broiler nutrition, pigment additives are often employed along with oils [180,181] to mitigate the spontaneous oxidative effects on fat deposits and to improve the carcass's oxidative stability [182]. Furthermore, dietary carotene addition shows controversial effects via vitaminic metabolism, showing antagonistic [183] and synergic action [175] and might have an opposite role as an antioxidant [184] and pro-oxidant factor [185], depending on factors such as dietary formulation (inclusion or addition). The antagonism between vitamin E accumulation and β-carotene was studied, and the results show that the presence of β-carotene in broiler breast meat tends to limit vitamin E accumulation [186]. However, the dietary addition of lycopene and vitamin E improves the broiler chicks' growth performance and tight meat oxidative stability and also presents a synergic benefic effect on thigh meat cholesterol content. In laying hens, diets include distinct amounts of corn and alfalfa meal, contributing to the content of native pigments in the diet [6]. Intensive rearing systems diets are low in native xanthophylls. Therefore, the egg yolk is often characterized by a pale-yellow color [6] due to rich amounts in barley, rice, or wheat that are supplemented with artificial pigments (β-apo-8ícarotenoic-acid-ethyl ester) to satisfy the range of color scores required by the European egg producers and to meet the consumer's expectations [187].

## 6. Conclusions

As a directed movement in the food and feed markets guided for more natural products, the demand for organic ingredients is rising. Feed formulation recipes using natural and organic additives are the new trend in livestock nutrition research, using not only active principles that affect vegetal but also microorganisms for valuable active bio compounds. Yeast pigments are outstanding sources of natural color, covering a wide range of nutri-

tional and medicinal properties. Both carotenoid yield and total carotenoid structure are important aspects that could be optimized depending on strategy, adopting strain genetic engineering and process development, and employing cheap organic substrates. Further studies are required to establish biological and chemical proprieties, and yeast carotenoid mechanisms, enhancing yeast carotenoid productivity, stability, and marketability as alternatives to classic synthetic pigments. Data generation concerning a highly productive yeast process involving scalability for large-scale adaptability to fermentation aspects (fermentation design and bioreactor types) is essential. Furthermore, studies regarding the effects of value-added yeast pigment additives on livestock health, productivity, and product quality are important in validating nutritional and medicinal potential. Not last, consumers' perceptions and preferences in buying animal products obtained with microbial pigment additives firmly increase the need for knowledge.

**Author Contributions:** Conceptualization, D.-M.G. and N.E.B.; methodology, M.U.-I.; validation, E.N.P. and N.E.B.; resources D.-M.G.; data curation, D.-M.G., and N.E.B.; writing—original draft preparation, D.-M.G. and N.E.B.; writing—review and editing, M.U.-I.; visualization, E.N.P.; supervision, N.E.B.; funding acquisition, E.N.P. All authors have read and agreed to the published version of the manuscript.

**Funding:** The article processing charge was supported by the University of Agronomic Sciences and Veterinary Medicine of Bucharest, the Faculty of Animal Productions Engineering and Management.

**Institutional Review Board Statement:** Not applicable.

**Data Availability Statement:** Not applicable.

**Conflicts of Interest:** The authors declare no conflict of interest.

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
