# Peer review of "Transforming Rhodotorula sp. Biomass to Active Biologic Compounds for Poultry Nutrition"

_agriculture, doi:10.3390/agriculture13061159_

Round 1

Reviewer 1 Report

Very interesting Journal Manuscript on an interesting and timely topic, technology, and the potential of yeasts as a source of color in poultry products.
Small individual errors would be:
L 62 - a.2% ? probably should be a space
L 107, 108 - you do not need a comma before the abbreviation in parentheses, you do not write it in the rest of the text.
L 151 - space out the 75 quote.
L 315 - what do you mean by "forage", you have feed, what are the forages in poultry feeding that you specifically state, I think you can leave out the "forage.

Author Response

Response to reviewer

 We would like to thank the reviewers for their time, consideration, and proper recommendations. We appreciate their thoughtful comments and efforts toward improving our manuscript.

In the following sections, we respond to the comments that were more specific to each referee.

REVIEWER 1 REPORT NO.1

MINOR COMMENTS:

L 62 - a.2%? probably should be a space.

Corrections have been made and presented between L62.

L 107, 108 - you do not need a comma before the abbreviation in parentheses, you do not write it in the rest of the text.

Corrections have been made; we deleted the commas before the abbreviations in the parentheses.

L 151 - space out the 75 quotes.

L 315 - what do you mean by "forage", you have a feed, what are the forages in poultry feeding that you specifically state, I think you can leave out the "forage.

We corrected them as suggestions (L151, 315), deleting the “forage”.

We improved the manuscript according to the suggestions.

Reviewer 2 Report

Dear editor of Agriculture

I hope all of you are always fine. Regarding the revision of the manuscript No. agriculture-2390696, titled “The biotechnology application for bioproducts: transforming Rhodotorula sp. biomass to active biologic compounds for poultry nutrition”. Really it is an interesting review, however, some comments should be replied.

Comments

1-    Please design a figure to summarize the major benefits of bioactive compounds of Rhodotorula sp. in poultry production.

2-    In table 2: you can merge both bioactive compounds, B carotine and  torularhodin, in the same row (Rhodotorula glutinis) with both references (47, 56).

3-    Table 4: please include more breeds of chickens (such as IR, Lohman, etc….), ducks and turkeys (the most important reared poultry species worldwide). Also, if you can, please indicate the difference in requirements of pigment additive (IU/kg fed) in dietary-fed formulas between white and brown breeds of chickens.

4-    References 147-151 need to be updated to the newest versions.

Author Response

Response to reviewer

 We would like to thank the reviewers for their time, consideration, and proper recommendations. We appreciate their thoughtful comments and efforts toward improving our manuscript.

In the following sections, we respond to the comments that were more specific to each referee.

REVIEWER 2 REPORT NO.1

COMMENTS:

1-    Please design a figure to summarize the major benefits of bioactive compounds of Rhodotorula sp. in poultry production.

We developed the design figure according to the suggestions, Figure 1, L105.

2-    In Table 2: you can merge both bioactive compounds, B carotene, and torularhodin, in the same row (Rhodotorula glutinis) with both references (47, 56).

We corrected as suggestions, merging both references for Rhodotorula glutinis.

3-    Table 4: please include more breeds of chickens (such as IR, Lohman, etc….), ducks, and turkeys (the most important reared poultry species worldwide). Also, if you can, please indicate the difference in requirements of pigment additive (IU/kg fed) in dietary-fed formulas between white and brown breeds of chickens.

We improved Table 4, according to the suggestions.

4-    References 147-151 need to be updated to the newest versions.

We updated the latest version of references 147-151.

We improved the manuscript according to the suggestions.

Reviewer 3 Report

Author given new idea but please short the title of paper

Add new updated references

Other is ok

ok

Author Response

Response to reviewer

 We would like to thank the reviewers for their time, consideration, and proper recommendations. We appreciate their thoughtful comments and efforts toward improving our manuscript.

In the following sections, we respond to the comments that were more specific to each referee.

REVIEWER 3 REPORT NO.1

COMMENTS:

Author given new idea but please shorten the title of the paper.

We have made changes to the title of the paper: Transforming Rhodotorula sp. biomass to active biologic com-pounds for poultry nutrition

Add new updated references

We have updated the references, for poultry nutrition requirements as previously suggested by the Reviewer No. 2 suggestions.

We improved the manuscript according to the suggestions.

Reviewer 4 Report

Major comments

In general, the section “2. Rhodotorula sp. general aspects” may be quite informative, however it contains information that may be beyond the scope of such a review and especially for the readers of the journal. It is advised to the authors to reduce the content of this section by 50%, retaining those aspects that may be important especially for those practitioners in the field such as animal scientists, veterinarians, etc.

L170-178: the authors will need to revise this paragraph. It lacks uniformity and the closing sentence “Colonization extent which causes natural degradation and spoilage” should be modified and better explained. It may also be questionable when referring to degradation and spoilage conditions, especially in relation to poultry industry products. Is it advised to use all these “High-sugar fruits and vegetables” which may not be in optimal condition for human consumption for example? Under such conditions other by-products which can generate undesirable effects may generate and potentially be harmful for the health and productivity of the birds.

L190-192: this statement may raise serious concerns and comments about the food chain and the effects on the health of birds and humans. The cannot be an unequivocal approach on this issue, as by other processes such as oxidation, other substances can be detected in high amounts in some by-products.

Minor comments

L85-95: could be reduced by 50% or more focusing on the most important aspects related to the topic of the review.

Table 2: remove “;” after “Aspartic protease – pepsin family;”. Similarly, with “Lipase (Lip-Y23) – low-temperature applications;”

Author Response

Response to reviewer

 We would like to thank the reviewers for their time, consideration, and proper recommendations. We appreciate their thoughtful comments and efforts toward improving our manuscript.

In the following sections, we respond to the comments that were more specific to each referee.

REVIEWER 4 REPORT NO.1

COMMENTS:

In general, the section “2. Rhodotorula sp. general aspects” may be quite informative, however, it contains information that may be beyond the scope of such a review and especially for the readers of the journal. It is advised to the authors to reduce the content of this section by 50%, retaining those aspects that may be important, especially for those practitioners in the field such as animal scientists, veterinarians, etc.

We kindly ask for your understanding considering this comment. We would like to keep the current format, due to the fact that the current manuscript is part of a doctoral thesis, biotechnology domain, focused on developing Rhodotorula yeast applications for poultry feed.

L170-178: the authors will need to revise this paragraph. It lacks uniformity and the closing sentence “Colonization extent which causes natural degradation and spoilage” should be modified and better explained. It may also be questionable when referring to degradation and spoilage conditions, especially in relation to poultry industry products. Is it advised to use all these “High-sugar fruits and vegetables” which may not be in optimal condition for human consumption for example? Under such conditions, other by-products which can generate undesirable effects may generate and potentially be harmful to the health and productivity of the birds.

We assume the writing error, and we corrected it. The full sentence is “High-sugar fruits and vegetables are natural habitats for yeasts such as Rhodotorula sp colonization extent which causes natural degradation and spoilage [85].” In this context, we highlight the potential of vegetal waste (fruit and vegetables) for obtaining yeast biomass, having in mind that Rhodotorula yeasts are vegetal natural inhabitants, thus natural fermenters.

L190-192: this statement may raise serious concerns and comments about the food chain and the effects on the health of birds and humans. They cannot be an unequivocal approach to this issue, as by other processes such as oxidation, other substances can be detected in high amounts in some by-products.

We strongly believe that recycling food waste as a potential source of nutrients, for yeasts, and furthermore processing yeasts biomass (having mandatory biomass inactivation) for livestock nutrition might represent a potential solution to the global actual challenge. Many studies previously published are presenting outstanding results by exploiting microorganisms, grown on waste substrates.

L85-95: could be reduced by 50% or more focusing on the most important aspects related to the topic of the review.

At L85-95 we have made modifications, as suggestions: The genus Rhodotorula sp. covers more than 165 species [34]. Morphologic, Rhodotorula sp. are polyphyletic shaped yeast [35], forming fast-growing colored colonies [36].

Table 2: remove “;” after “Aspartic protease – pepsin family;”. Similarly, with “Lipase (Lip-Y23) – low-temperature applications;”

We removed the “;”, after “Aspartic protease – pepsin family” and “Lipase (Lip-Y23) – low-temperature applications”.

We improved the manuscript according to the suggestions.

Round 2

Reviewer 4 Report

The authors have not complied with most of the suggestions and modifications proposed. Limitations were not addressed.

Author Response

Response to Academic Editor and Reviewer no 4

We would like to thank the academic editor and the reviewer for their time, and consideration. We appreciate their efforts towards improving our current manuscript.

In the following sections, we respond to the suggestions.

ACADEMIC EDITOR AND REVIEWER 4 REPORT NO.2 COMMENTS:

In general, the section “2. Rhodotorula sp. general aspects” may be quite informative, however, it contains information that may be beyond the scope of such a review and especially for the readers of the journal. It is advised to the authors to reduce the content of this section by 50%, retaining those aspects that may be important, especially for those practitioners in the field such as animal scientists, veterinarians, etc.

We merged section 2, and we reduced it by around 40%, retaining general aspects, and those that might be important for the animal science-related fields.

L170-178: the authors will need to revise this paragraph. It lacks uniformity and the closing sentence “Colonization extent which causes natural degradation and spoilage” should be modified and better explained. It may also be questionable when referring to degradation and spoilage conditions, especially in relation to poultry industry products. Is it advised to use all these “High-sugar fruits and vegetables” which may not be in optimal condition for human consumption for example? Under such conditions, other by-products which can generate undesirable effects may generate and potentially be harmful to the health and productivity of the birds.

We revised the paragraph, and we deleted the last sentence.

L190-192: this statement may raise serious concerns and comments about the food chain and the effects on the health of birds and humans. They cannot be an unequivocal approach to this issue, as by other processes such as oxidation, other substances can be detected in high amounts in some by-products.

We improved our statement.

L85-95: could be reduced by 50% or more focusing on the most important aspects related to the topic of the review.

At L85-95 we have made modifications and reduced it by 50%.

Table 2: remove “;” after “Aspartic protease – pepsin family;”. Similarly, with “Lipase (Lip-Y23) – low-temperature applications;”

We removed the “;”, after “Aspartic protease – pepsin family” and “Lipase (Lip-Y23) – low-temperature applications”.

We improved the manuscript according to the suggestions.